# Factors associated with hemolysis during extracorporeal membrane oxygenation (ECMO)—Comparison of VA- versus VV ECMO

Hannah Appelt[1], Alois Philipp[1], Thomas Mueller[2], Maik Foltan[1], Matthias Lubnow[2], Dirk Lunz[3], Florian Zeman[4], Karla Lehle[1] *

**1** Department of Cardiothoracic Surgery, University Hospital Regensburg, Regensburg, Germany, **2** Department of Internal Medicine II, University Hospital Regensburg, Regensburg, Germany, **3** Department of Anesthesiology, University Hospital Regensburg, Regensburg, Germany, **4** Center for Clinical Studies, University Hospital Regensburg, Regensburg, Germany

* Karla.Lehle@ukr.de

**Data Availability Statement:** All relevant data are within the manuscript and its Supporting Information files.

## Abstract

Venovenous (VV) and venoarterial (VA) extracorporeal membrane oxygenation (ECMO) are effective support modalities to treat critically ill patients. ECMO-associated hemolysis remains a serious complication. The aim was to disclose similarities and differences in VA- and VV ECMO-associated hemolysis. This is a retrospective single-center analysis (January 2012 to September 2018) including 1,063 adult consecutive patients (VA, n = 606; VV, n = 457). Severe hemolysis (free plasma hemoglobin, fHb > 500 mg/l) during therapy occurred in 4% (VA) and 2% (VV) (p≤0.001). VV ECMO showed significantly more hemolysis by pump head thrombosis (PHT) compared to VA ECMO (9% vs. 2%; p≤0.001). Pre-treatments (ECPR, cardiac surgery) of patients who required VA ECMO caused high fHb pre levels which aggravates the proof of ECMO-induced hemolysis (median (interquartile range), VA: fHb pre: 225.0 (89.3–458.0); VV: fHb pre: 72.0 (42.0–138.0); p≤0.001). The survival rate to discharge from hospital differed depending on ECMO type (40% (VA) vs. 63% (VV); p≤0.001). Hemolysis was dominant in VA ECMO patients, mainly caused by different indications and not by the ECMO support itself. PHT was the most severe form of ECMO-induced hemolysis that occurs in both therapies with low frequency, but more commonly in VV ECMO due to prolonged support time.

## Introduction

Technical-induced hemolysis remains a common and critical complication during extracorporeal membrane oxygenation (ECMO) contributing to a variety of adverse events, and consequently affecting patient survival and quality of life. Reported incidence was between 5 and 18% [1–9]. Hemolysis is a mechanical damage of red blood cells (RBCs) induced by excessive high shear stress, like blood pumps, cannulas, which is manifested by hemoglobin released from ruptured, overstretched, overheated or prematurely aged RBCs. Free plasma hemoglobin (fHb) is cytotoxic resulting in tissue hypoxia and cell death [4]. FHb scavenges nitric oxide,

**Funding:** The authors received no specific funding for this work.

**Competing interests:** The authors have declared that no competing interests exist.

leading to inappropriate vasoconstriction, endothelial dysfunction and platelet aggregation [5]. Consequently, severe complications such as renal dysfunction or multiple organ failure may emerge [1,8,10]. Therefore, prompt identification of technical-induced hemolysis is essential.

Comparative studies on hemolysis during adult venoarterial (VA) and venovenous (VV) ECMO are scarce. Mostly, patients treated with VA- and/or VV ECMO were included in studies for a large cohort analysis [9]. This is to our knowledge the first study that describes the differences and similarities of VA- and VV ECMO in relation to hemolysis. The prevalence of hemolysis and various factors that may lead to an increased fHb value were analyzed. FHb was used as a sensitive marker for hemolysis [1,11,12].

## Methods

### Study design

This is a retrospective single-center analysis on prospectively collected data (Regensburg ECMO Registry, medical records collected from January 2012 to September 2018) from consecutive patients that were treated with VA ECMO (n = 606) and VV ECMO (n = 457). The Ethics Committee of the University Regensburg approved the study protocol (vote #17-568-104). The need for informed consent was waived by the Ethics Committee, as all devices are approved for clinical use, no personalized data (fully anonymization) and only routine laboratory parameters were used.

### Study population

ECMO management, indications and limitations have been described previously [1,13,14]. Our ECMO center mainly used four different ECMO systems. Table 1 listed all systems that were used first for therapy. Patients younger than 18 years were excluded.

### Data collection and analysis

Data of this study were acquired from the Regensburg ECMO database, in which prospective physical and laboratory parameters, ECMO management data and outcome of ECMO patients are collected. Plasma free hemoglobin (fHb) levels were used as a marker for hemolysis [1,11,12]. The quantitative measurements of the fHb values from the patient's blood were performed with a Dimension Vista® 1500 Clinical Chemical Analyzer (Siemens Healthcare Diagnostics GmbH, Eschborn, Germany). In order to avoid errors, a new measurement was

**Table 1. Distribution of the different ECMO systems.**

| System | Pump | Oxygenator | VA ECMO [n; %] | VV ECMO [n; %] |
|---|---|---|---|---|
| | | | 606 | 457 |
| **Cardiohelp HLS** | Cardiohelp | Cardiohelp 5.0 / 7.0 | 347; 57 | 136; 30 |
| **PLS** | Rotaflow | Quadrox D | 96; 16 | 74; 16 |
| **DP3 system** | DP3 | Hilite LT 7000 | 54; 9 | 135; 30 |
| **Life-Box** | Revolution | ECC.O 5 | 75; 12 | 92; 20 |
| **Others** | Rotaflow | MECC Quadrox Softline | 34; 6 | 5; 1.1 |
| | HemoLung | HemoLung | 0; 0 | 3; 0.7 |
| | DP3 | iLA activve | 0; 0 | 12; 2.6 |

Cardiohelp HLS, PLS and MECC: Getinge / Maquet GmbH, Rastatt, Germany; DP3 system and iLA-activve: Fresenius / Xenios AG, Heilbronn, Germany; Life-Box: Sorin Group / Liva Nova, Milan, Italy; HemoLung: ALung Technologies, Pittsburgh, USA.

performed for conspicuously high values or from a doubling of the values within one day. FHb values > 500 mg/l were evaluated as clinically conspicuous and critical. FHb values before ECMO implantation (fHb pre) were of special interest. The concentrations were recorded daily until end of therapy. High fHb pre levels were used to identify subgroups of patients with a preexisting disease or intervention (such as cardiopulmonary resuscitation, CPR). FHb levels on the 1$^{st}$ day on ECMO were used to identify the effect of ECMO and ECMO circuit compounds (such as blood pump or cannulation) on the development of hemolysis. A total of 8,617 fHb values (from the time before ECMO support until its end) were included in the analysis. In addition, lactate dehydrogenase (LDH), bilirubin, C-reactive protein (CRP), tumor necrosis factor alpha (TNF-α), interleukin-6 (IL-6) and platelets were analyzed as laboratory parameters.

## Statistics

Statistical analysis was done using SigmaStat 3.5 (SYSTAT Software, San Jose, CA, USA). The used data sets are located in a supplementary file (dataset.xlsx). Continuous variables were shown as median (interquartile range, IQR), categorical variables were expressed as number (percentage). To compare e.g. the fHb values before and after ECMO initiation or during ECMO treatment, the paired values were checked for normal distribution using the Shapiro-Wilk test. If data were not normally distributed, the Wilcoxon non-parametric sign rank test was used. If analyses of unpaired values were necessary, statistical correlations were determined using the Mann-Whitney-U test. The Chi-square test was used if nominal distributed parameters were to be tested for correlation. The Z-test was used to compare proportions. P-value ≤ 0.05 was considered the threshold of statistical significance.

## Results

### Study population

While the majority of VA ECMOs used the Cardiohelp HLS system (57%), VV ECMO used both Cardiohelp HLS and DP3 system (each, 30%) equally (Table 1). The indications for the use of VA- or VV ECMO are different. This led to differences in patient characteristics and initial laboratory values (Table 2). Patients of both groups did not differ in gender, SOFA score, IL-6 and platelets. In addition, there was no difference in initial norepinephrine doses and the proportion of patients with acute renal failure (ARF), defined as the need for renal replacement therapy.

However, VA ECMO patients were older in age with a lower BMI and higher initial levels for fHb and LDH. In contrast, inflammatory data (CRP, TNF-α) and Bilirubin, a product of hemoglobin degradation and indicator for liver function, were significantly elevated in patients before VV ECMO implantation. Furthermore, the patients who required VA ECMO showed significantly higher aPTTs compared to patients with VV ECMO (Table 2). Indications for VA ECMO were ECPR (48%), cardiogenic shock (CS) (42%) and no weaning from cardio-pulmonary bypass (NWCPB) (10%). The main indication for VV ECMO was pulmonary acute lung failure (ALF) (71%). As expected from a former study [11] the requirement of CPR before ECMO caused hemolysis. Table 2 shows similar results for patients with ECPR and NWCPB. Patients with CS showed significantly lower fHb pre levels compared to ECPR and NWCPB (p≤0.001). Furthermore, patients with extrapulmonary ALF who required VV ECMO presented significantly elevated levels for fHb pre compared to patients with pulmonary ALF (p = 0.002).

**Table 2. Patient characteristics and preoperative laboratory data.**

| | VA ECMO | VV ECMO | p-value |
|---|---|---|---|
| Patients [n] | 606 | 457 | - |
| Females [n; %] | 170; 28 | 141; 31 | p = 0.092 |
| Age [years] | 60.4 (51.3–68.4) | 54.8 (44.1–63.7) | p≤0.001 |
| SOFA Score | 12.0 (9.0–14.0) | 12.0 (9.0–15.0) | p = 0.815 |
| BMI [kg * m$^{-2}$] | 26.7 (24.2–30.1) | 27.7 (24.2–33.1) | p≤0.001 |
| ARF [n; %] | 102; 17 | 96; 21 | p = 0.099 |
| NE [µg/kg/min] | 0.30 (0.14–0.64) | 0.31 (0.13–0.60) | p = 0.950 |
| aPTT [sec] | 53 (37–103) | 42 (35–53) | p≤0.001 |
| FHb pre [mg/l] | 225 (89–458) | 72 (42–138) | p≤0.001 |
| LDH pre [U/l] | 483.0 (281.0–860.0) | 397.5 (265.5–637.8) | p≤0.001 |
| Bilirubin pre [mg/dl] | 0.7 (0.4–1.3) | 0.8 (0.5–1.9) | p≤0.001 |
| CRP pre [mg/l] | 17.5 (4.0–80.3) | 140.0 (42.5–250.0) | p≤0.001 |
| TNF-α pre [pg/ml] | 15.0 (10.0–23.0) | 24.5 (14.0–47.0) | p≤0.001 |
| IL-6 pre [pg/ml] | 409.0 (149.0–1520.0) | 464.0 (100.5–4620.0) | p = 0.433 |
| Platelets pre [/nl] | 176.0 (123.0–240.5) | 175.0 (109.0–253.0) | p = 0.823 |
| ECMO indication: | | | |
| ECPR [n; %] | 293; 48 | - | - |
| fHb pre [mg/l] | 289 (141–552) | | |
| CS [n; %] | 254; 42 | - | - |
| fHb pre [mg/l] | 129 (61–286) [a] | | |
| NWCPB [n; %] | 59; 10 | - | - |
| fHb pre [mg/l] | 282 (113–602) | | |
| Pulm. ALF [n; %] | - | 327; 71 | - |
| fHb pre [mg/l] | | 65 (40–119) | |
| Extrapulm. ALF [n; %] | - | 131; 29 | - |
| fHb pre [mg/l] | | 100 (50–189) [b] | |

Data are shown as median (interquartile range); except for patient number, female gender, and ECMO indication (n; %). ALF, acute lung failure; BMI, Body mass index; ARF, acute renal failure; NE, Norepinephrine; aPTT, activated partial thromboplastin time; CRP, C-reactive protein; CS, cardiogenic shock: ECPR: extracorporeal cardiopulmonary resuscitation; fHb: free plasma hemoglobin; IL-6, interleukin-6; LDH, lactate dehydrogenase; SOFA, Sequential Organ Failure Assessment); TNF-α, tumor necrosis factor; NWCPB, no weaning from cardio-pulmonary bypass; ALF, acute lung failure. Pulmonary ALF: bacterial, viral, fungal, aspiration pneumonia, ALF not post trauma, other pathologies (e.g. pulmonary fibrosis, near drowning). Extrapulmonary ALF: ALF post trauma, surgery, chemotherapy.

[a], fHb pre from CS was significantly lower compared to ECPR and NWCPB, p≤0.001;

[b], fHb pre from extrapulmonary ALF was significantly higher than from pulmonary ALF, p = 0.002.

## Impact of cannulation on the development of hemolysis

Small cannulas and high ECMO blood flow may induce hemolysis [15]. VA- and VV ECMO used different initial cannulation strategies (Table 3).

While VV ECMO only used a minimally invasive method (peripheral percutaneous cannulation), 16% of VA ECMO patients additionally required central and peripheral surgical cannulation. Double lumen cannulas (DLCs) were used only for VV ECMO (22%).

FHb on the 1st day on ECMO was used to verify hemolysis induction due to different cannulation strategies. Neither central, nor peripheral surgical or peripheral percutaneous cannulation of VA ECMO patients affected fHb levels (Table 3, p = 0.973). In addition, usage of

**Table 3. Cannulation strategies and effect on hemolysis.**

|  | VA ECMO | VV ECMO | p-value |
|---|---|---|---|
| Patients [n] | 606 | 457 | - |
| **Cannulation technique** |  |  |  |
| central [n; %] | 57; 9 | - | - |
| fHb 1$^{st}$ day [mg/l] | 89 (46–309) |  |  |
| peripheral surgical [n; %] | 38; 6 | - | - |
| fHb 1$^{st}$ day [mg/l] | 93 (44–213) |  |  |
| peripheral percutaneous [n; %] | 511; 84 | 457; 100 |  |
| fHb 1$^{st}$ day [mg/l] | 81 (51–163) | 72 (42–138) | 0.973 |
| **Cannula type** |  |  |  |
| SLC [n; %] | 606; 100 | 357; 78 |  |
| fHb, 1$^{st}$ day [mg/l] | 82 (49–172) | 58 (40–102) | p≤0.001 |
| DLC [n; %] | - | 100; 22 | - |
| fHb, 1$^{st}$ day [mg/l] |  | 67 (40–107) |  |

Data are median (interquartile range). FHb, free plasma hemoglobin; DLC, double lumen cannula; SLC, single lumen cannula.

DLCs during VV ECMO did not induce hemolysis compared to single lumen cannulas (SLCs, p = 0.358).

Small-sized inflow cannulas (17 Fr) at a blood flow of ≤ 2.5 l/min (low flow) and > 3.0 l/min (high flow) did not induce hemolysis during VV ECMO [16]. Respective data for VA ECMO failed so far. VA ECMO patients were mainly supported with 15 Fr (50%) and 17 Fr (32%) cannulas. Despite differences in the inner diameter of these SLCs (15 Fr, 4.29 mm; 17 Fr, 4.85 mm) and a higher blood flow requirement for the 17 Fr SLCs (resulting in a significantly lower flow velocity), the fHb levels on the 1$^{st}$ and 2$^{nd}$ day on ECMO were comparable between both cannula types (Table 4). The downregulation of the blood flow from 1$^{st}$ to 2$^{nd}$ day was accompanied by a significant decrease in fHb levels. The support time was comparable for the different SLCs.

**Table 4. Effect of 15 Fr and 17 Fr inflow cannula (VA ECMO) on hemolysis.**

|  | 15 Fr | 17 Fr | p-value |
|---|---|---|---|
| Patients [n; %] | 306; 50 | 194; 32 | - |
| fHb, 1$^{st}$ day [mg/l] | 78 (50–143) | 85 (51–209) | p = 0.142 |
| fHb, 2$^{nd}$ day [mg/l] | 63 (43–97) | 59 (43–125) | p = 0.715 |
| p-value, 1$^{st}$ vs. 2$^{nd}$ day | p = 0.002 | p≤0.001 |  |
| BF, 1$^{st}$ day [l/min] | 2.9 (2.3–3.3) | 3.3 (2.8–3.9) | p≤0.001 |
| BF, 2$^{nd}$ day [l/min] | 2.7 (2.2–3.3) | 3.1 (2.5–3.7) | p = 0.002 |
| p-value, 1$^{st}$ vs. 2$^{nd}$ day | p≤0.001 | p≤0.001 |  |
| FV, 1$^{st}$ day [cm/s] | 20.1 (15.9–22.8) | 17.8 (15.1–21.1) | p≤0.001 |
| FV, 2$^{nd}$ day [cm/s] | 18.7 (15.2–22.8) | 16.8 (13.5–20.0) | p≤0.001 |
| p-value, 1$^{st}$ vs. 2$^{nd}$ day | p≤0.001 | ≤0.001 |  |
| ECMO time [days] | 3.0 (2.0–6.0) | 3.0 (1.8–6.0) | p = 0.932 |

Data are median (interquartile range) of all patients with 15 Fr and 17 Fr cannulas. fHb, free plasma hemoglobin; Fr, French; BF, blood flow; FV, flow velocity. FV was calculated by dividing blood flow (Q, cm$^3$/s) through the cross-sectional area of the cannula (A, cm$^2$), FV = Q/A (cm/s)

Including only patients with fHb pre $\leq$ 100 mg/l and without PHT (Fig 1), there was also no effect of cannula size on the induction of hemolysis (1st, 2nd day, Fig 1A). As expected (see above), 17 Fr cannulas enabled significantly higher blood flow compared to 15 Fr cannulas (independent of the day on ECMO, Fig 1B). However, coding for high ($\geq$ 3.0 l/min) and low ($\leq$ 2.5 l/min) blood flow requirements resulted in a significant hemolysis for 15 Fr cannulas at high blood flow compared to 17 Fr cannulas ((Fig 1C). Furthermore, at high blood flow, the fHb levels of 15 Fr cannulas were significantly lower compared to 17 Fr cannulas (p = 0.027).

Furthermore, ECPR patients that presented high fHb pre levels (S1 Table) were of special interest regarding cannula effects. Within one day on ECMO, high fHb pre levels decreased significantly independent of cannula size (15 Fr, decrease of 34%, p$\leq$0.001; 17 Fr, decrease of 22%, p = 0.003). However, there was no difference of fHb levels on the 1st and 2nd day on ECMO comparing 15 Fr and 17 Fr cannulas (p = 0.853 and p = 0.729, respectively). The differences in blood flow, flow velocity and ECMO time described in Table 4 also apply to this group of patients.

## Distribution of pump systems in the investigated patient population

The effect of blood pumps, in particular centrifugal pumps, on hemolysis is debatable [17,18]. As shown in Table 5, VA ECMO patients were more frequently supplied with Cardiohelp pumps (Cardiohelp HLS system), while VV ECMO patients used more DP3 pumps.

Nevertheless, independent of ECMO type, the single pump type had no impact on the development of hemolysis (VA: 1st day, p = 0.077; VV: 1st day, p = 0.072). Indeed, VA ECMO patients presented significantly higher levels of fHb pre and on 1st day compared to VV ECMO patients. However, none of the used pumps induced hemolysis. Instead, the concentrations decreased significantly within one day on VA ECMO (Table 5). VV ECMO showed a similar trend, but the decrease was less pronounced (Table 5).

## Hemolysis and pump head thrombosis

Hemolysis is a critical complication during ECMO therapy [1,3,5]. At our ECMO center, the levels of fHb were determined routinely every day. The frequency of complete measurements was 95% and 97% for VA- and VV ECMO, respectively (Table 6).

The cumulative ECMO time as well as the median ECMO time per patient was longer for VV ECMO. Fig 2 illustrates all measured fHb levels from each patient during its ECMO support.

In particular, the amount of critical fHb values ($>$ 500 mg/l) was significantly lower for VV ECMO (VA, 4%; VV, 2%, p$\leq$0.001). In addition, the proportion of patients with at least one critical fHb value was higher for VA ECMO (VA, 12%; VV, 10%; p$\leq$0.001). Furthermore, the occurrence of severe hemolysis–in particular of PHT–was significantly more frequent in VV ECMO patients (VA, 2%; VV, 9%, p$\leq$0.001). Time to onset of PHT was prolonged for VV ECMO patients. Pump head thrombosis is an acute emergency event and is defined as a rapid and substantial increase in fHb ($>$ 300 mg/l) accompanied mostly by a decrease in platelets and an abnormal noise/vibration of pump head. The pump head has to be changed. The levels of fHb decreased and platelets normalized after immediate change of the circuit [1]. In addition, the removed pump head presented visible clots.

There was no prevalence for the appearance of a PHT regarding the different pump systems (S2 Table). There was no difference in distribution of pumps with PHT for the ECMO types (p = 0.914). However, the incidence of a PHT was significantly elevated for Cardiohelp during VV ECMO (VA, 7/347, 2% vs. VV, 14/136, 10%, p$\leq$0.001). In contrast, the incidence of PHT was not different for the other pump systems comparing VA and VV data (Rotaflow, 1/96 vs.

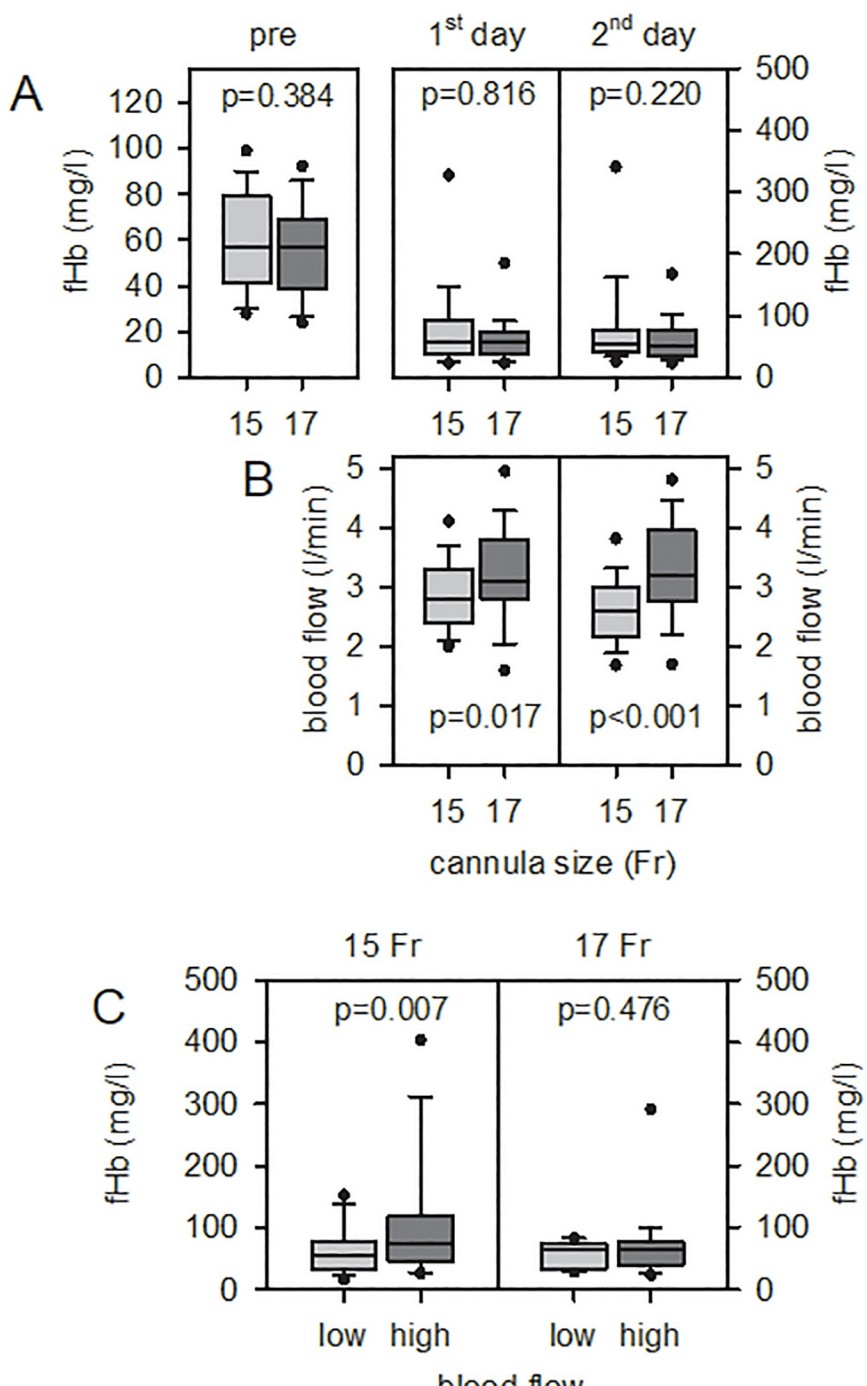

**Fig 1. Effect of cannula size and blood flow on hemolysis induction regarding VA ECMO patients with fHb pre ≤ 100 mg/l (without pump head thrombosis).** 15 Fr n = 86, 17 Fr n = 36. (A) Cannula size (Fr, French) had no effect on fHb levels on 1st and 2nd day on ECMO. (B) 17 Fr cannulas required a significantly higher blood flow compared to 15 Fr cannulas. (C) High blood flow (≥ 3 l/min) within 15 Fr cannulas induced significantly higher fHb levels compared to low blood flow (≤ 2.5 l/min). The median is shown as a black line in the box. The 25% or 75% quantile represents the lower or upper limit of the box. The smallest and largest observation is shown as whiskers, extreme values as circles.

**Table 5. Effect of pump type during VA- and VV ECMO on hemolysis.**

| System | Cardiohelp HLS | PLS | DP3 system | Life-Box | Others |
|---|---|---|---|---|---|
| **Pump** | Cardiohelp | Rotaflow | DP3 | Revolution | Others |
| **VA ECMO** | | | | | |
| **n; %** | 347; 57 | 96; 16 | 54; 9 | 75; 12 | 34; 6 |
| **fHb pre [mg/l]** | 227 (100–450) | 233 (101–583) | 129 (67–276) | 246 (79–412) | 275 (84–596) |
| **fHb, 1st day [mg/l]** | 74 (46–154) | 80 (49–243) | 93 (48–167) | 101 (56–264) | 107 (57–292) |
| **p-values (pre vs. 1st day)** | $p \leq 0.001$ | $p \leq 0.001$ | $p = 0.017$ | $p = 0.071$ | $p = 0.003$ |
| **VV ECMO** | | | | | |
| **n; %** | 136; 30 | 74; 16 | 135; 29 | 92; 20 | 20; 5 |
| **fHb pre [mg/l]** | 85 (46–145) | 55 (36–145) | 69 (45–135) | 65 (37–133) | 73 (49–125) |
| **fHb, 1st day [mg/l]** | 64 (44–109) | 56 (37–99) | 62 (39–110) | 53 (35–76) | 51 (32–98) |
| **p-values (pre vs. 1st day)** | $p = 0.003$ | $p = 0.034$ | $p = 0.020$ | $p = 0.002$ | $p = 0.225$ |
| **p-values (VA vs. VV)** | | | | | |
| **pump** | $p \leq 0.001$ | $p = 0.833$ | $p = 0.006$ | $p = 0.239$ | $p = 0.647$ |
| **fHb pre** | $p \leq 0.001$ | $p \leq 0.001$ | $p = 0.002$ | $p \leq 0.001$ | $p = 0.005$ |
| **fHb, 1st day** | $p = 0.092$ | $p \leq 0.001$ | $p = 0.027$ | $p \leq 0.001$ | $p = 0.032$ |

Data are median (interquartile range); except for patient number. Cardiohelp and Rotaflow: Getinge / Maquet GmbH, Rastatt, Germany; DP3: Fresenius / Xenios AG, Heilbronn, Germany; Revolution: Sorin Group / Liva Nova, Milan, Italy. Others: Various pumps (e.g. HemoLung: ALung Technologies, Pittsburgh, USA). Statistics compared respective data from VA- and VV ECMO.

4/74 vs., p = 0.226; DP3, 2/54 vs. 8/135, p = 0.797; Revolution, 4/75 vs.13/92, p = 0.107; others, 1/34 vs. 2/20, p = 0.548).

## Blood products, ARF and outcome

During ECMO support, the consumption of blood products (RBC, FFP, PC) was significantly higher for VA ECMO support (Table 7). However, the proportion of patients that required RBC transfusion was not different between VA and VV ECMO therapy (VA, n = 422, 70%; VV: n = 305, 67%; p = 0.348). The incidence of acute renal failure was comparable between VA and VV ECMO (26% and 21%, respectively). Survival to hospital discharge was 40% vs. 63% ($p \leq 0.001$) for patients treated with VA- and VV ECMO, respectively (Table 7).

**Table 6. Frequency of hemolysis and pump head thrombosis.**

| 1st day-end | VA ECMO | VV ECMO | p-value |
|---|---|---|---|
| **Patients [n]** | 606 | 457 | - |
| **Cumulative ECMO time [days]** | 3010 | 4969 | - |
| **Cumulative fHb values [n]** | 2864 | 4832 | - |
| **Frequency [%]** | 95 | 97 | - |
| **ECMO time per patient [days; median (IQR)]** | 4 (2–7) | 8 (6–14) | $p \leq 0.001$ |
| **FHb values per patient [n; median (IQR)]** | 3 (1–6) | 8 (5–13) | $p \leq 0.001$ |
| **FHb values > 500 mg/l [n; %]** | 119; 4 | 86; 2 | $p \leq 0.001$ |
| **Patients with fHb > 500 mg/l [n; %]** | 72; 12 | 47; 10 | $p \leq 0.001$ |
| **PHT [n; %]** | 15; 2 | 41; 9 | $p \leq 0.001$ |
| **Time of PHT [days, median (IQR)]** | 5.0 (3.0–6.5) | 9.0 (7.0–13.0) | $p \leq 0.001$ |

FHb: free plasma hemoglobin; IQR: interquartile range; PHT: pump head thrombosis.

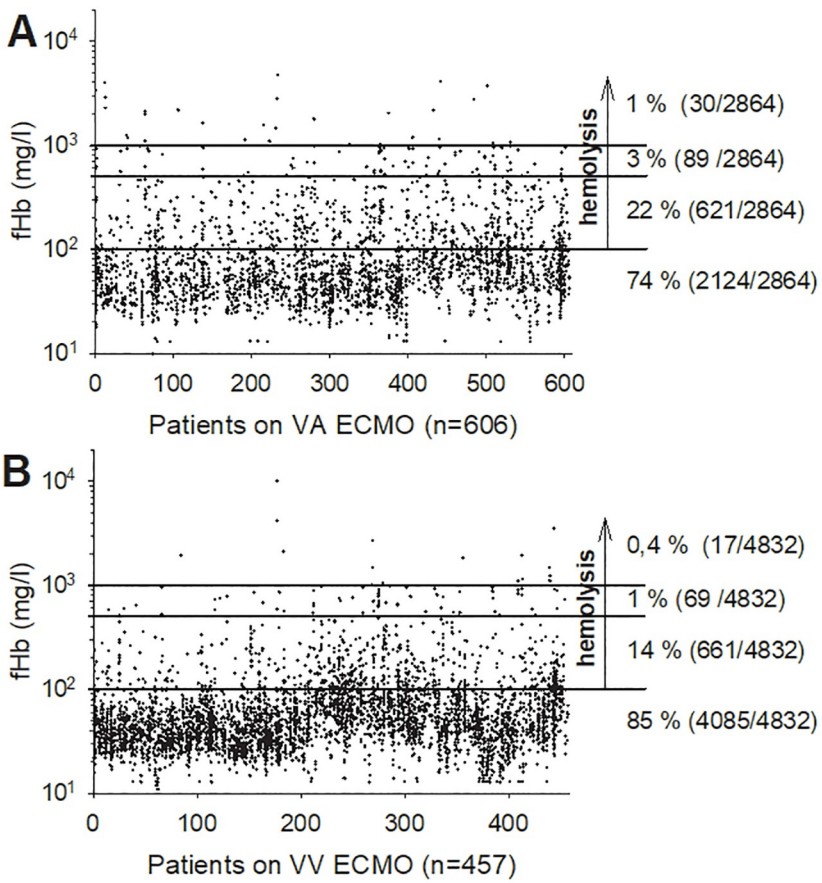

**Fig 2. All fHb measurements from 1st day till end of VA- or VV ECMO support (A: VA n = 606, B: VV n = 457).**
The lines divide the graph into values ≤ 100 mg/l ($10^2$), 101–500 mg/l, 501–1000 mg/l and > 1,000 mg/l ($10^3$) which
indicates a rising degree of hemolysis. The arrow shows all fHb above 100 mg/l that are suspected of hemolysis: 26%
(VA) vs. 15% (VV). Of particular importance were fHb values above 500 mg/l as critical hemolysis markers [2,9]: 1%
(VA) vs. 0.4% (VV).

**Table 7. Transfusion, acute renal failure and outcome.**

|  | VA ECMO | VV ECMO | p-value |
|---|---|---|---|
|  | 606 | 457 | - |
| **RBC/days ECMO** | 0.66 (0.00–1.77) | 0.26 (0.00–0.67) | p≤0.001 |
| **FFP/days ECMO** | 0.00 (0.00–1.00) | 0.00 (0.00–0.00) | p≤0.001 |
| **PC/days ECMO** | 0.00 (0.00–0.27) | 0.00 (0.00–0.00) | p≤0.001 |
| **ARF during ECMO [n; %]** | 159; 26 | 97; 21 | p = 0.069 |
| **Survival to hospital discharge [n; %]** | 241; 40 | 288; 63 | p≤0.001 |

Data are median (interquartile range) except for acute renal failure (ARF) on ECMO and survival rate. RBC, red
blood cells; FFP, fresh frozen plasma (1 FFP contains 230 ml plasma); PC, platelet concentrate (1 PC contains 250 ml
and 2–4 x $10^{11}$ platelets).

## Discussion

This study presents differences and similarities of hemolysis on a large cohort of adult patient
supported with VA- and VV ECMO. Patients with ECPR and NWCPB requiring VA ECMO

support showed highest levels of fHb even before ECMO implantation. Nevertheless, neither cannula nor pump type induced hemolysis during VA- and VV ECMO support. However, the frequency of high fHb values was significantly higher in the cohort of VA ECMO compared to VV ECMO patients. In addition, the incidence of PHT was significantly higher for VV ECMO compared to VA ECMO support.

The indication for VA- or VV ECMO depended on the supported organ: either cardiac and/or respiratory support. Therefore, the patient populations of both ECMO types differ due to their underlying diseases. Nevertheless, in many studies both patient collectives were always mixed, as the patient numbers at the individual centers were limited [9]. In the present study, both patient groups showed comparable SOFA scores, but VA ECMO patients were older and presented higher levels of fHb and LDH. The latter is a clear sign of cell destruction according to their indication. In contrast, increased levels of inflammatory parameters in septic patients with VV ECMO did not result in increased hemolysis [19]. The introduced high levels of fHb pre mainly came from ECPR and NWCPB procedures. In a former study, it was shown that CPR with chest compression caused hemolysis [11]. Furthermore, extensive surgery such as cardiac surgery with a pronounced need for RBC transfusions induces elevated fHb levels [20,21]. However, transfusion requirements before ECMO were not included in the ECMO database. In addition, metabolic disorders [22], hypoxia [23] due to cardiac arrest, various diseases [4], bleedings [20] and other large surgical interventions may increase RBC transfusions with increased levels of fHb.

High fHb pre levels complicated investigations of the primary impact of ECMO on red blood cell destruction. Therefore, former studies only used cases with fHb pre levels ≤ 100 mg/l to demonstrate that neither VA- nor VV ECMO aggravated hemolysis [11,16]. Regarding all patients–in particular, patients with high initial hemolysis–the implantation of an ECMO system resulted in a significant decrease in fHb levels. This effect was particularly noticeable in VA ECMO patients independent of cannulation strategy and pump type. ECMO-induced hemolysis apparently plays a significant role only in 15 Fr cannulas under high blood flow (≥ 3 l/min). About 50% of VA ECMO patients got a 15 Fr cannula. The size of the cannula depended on the requirement of anticipated tissue oxygenation and the degree of cardiac support provided [24]. An unexpectedly elevated demand required higher flow rates. The subsequent installation of a larger cannula is avoided because of the high risk for the patient (e.g. bleeding, infection). Obviously, extracorporeal support of circulation with restoration of tissue supply normalized RBC damage [11]. Cardiac arrest and cardiogenic shock impaired organ perfusion (e.g. stasis in spleen or liver). Obviously, a successful and rapid restoration of adequate cardiac output will eliminate locally harmful/toxic concentration of fHb in tissue or increase its degradation.

Nevertheless, hemolysis remains a critical complication during ECMO support. In literature, the frequency of hemolysis ranged between 5 and 18% [1–9]. At our ECMO center, severe hemolysis with fHb levels > 500 mg/l dominated in VA ECMO patients. The proportion of abnormal high fHb levels on ECMO was significantly higher for VA- (4%) compared to VV ECMO treatment (2%). In addition, the proportion of patients that presented high fHb values during ECMO was significantly higher for VA ECMO. These patients required significantly higher amounts of blood products (by a factor of 2.5 more RBCs) and tended to a higher proportion of ARF during ECMO. Another retrospective study from [25] demonstrated that RBC transfusion during ECMO was associated with severe complications (thromboembolic complications, sepsis, hemolysis) and ARF. Red cell damage during storage and the potential harmful consequences after transfusion are discussed in literature–in particular in massive transfusion scenarios [26].

However, only 13% of the high fHb levels were explainable by a life-threatening PHT. A PHT is always an acute event and requires a prompt circuit change with all the associated

complications and risks (such as bleedings, infections). However, within one day, fHb levels normalized. Interestingly, the incidence of a PHT was significantly higher for VV ECMO patients (9% vs. 2%). Furthermore, the time to PHT was by a factor of two significantly longer for VV ECMO. The differences described above may be due to the extended ECMO time. A prolonged use of cannulas in situ may lead to increased risk of clot formation on the cannula tip that can detach and aspirate in the pump head. In addition, increased inflammation within VV ECMO patients was also associated with hypercoagulability that may affect the incidence of PHT [27]. Furthermore, anticoagulation regimen was different between VA and VV ECMO (aPTT target values: VA: 50–60 sec; VV: 45–50 sec). In particular, cardiothoracic patients with worsened ventricular function and intracardiac stasis required higher anticoagulation to prevent a hypercoagulable state and the development of thrombosis. Finally, reasons for the different incidences of PHT cannot be clarified.

Another point of view is the pump-type specific induction of hemolysis [12,17,18,28,29]. Even the new generation of centrifugal pumps showed large differences in blood cell damage [29]. In the present study, there was no prevalence for appearance of a PHT regarding the different pump systems. However, the most used system (Cardiohelp HLS) showed a significantly higher proportion of PHT during VV ECMO support compared to VA ECMO (10% vs. 2%) if one compares the Cardiohelp HLS fraction with PHT with the total number of Cardiohelp HLS within the corresponding collective. Again, longer support time of VV ECMO as well as different states of coagulation may induce hemolysis. However, all other systems showed no difference in induction of hemolysis comparing the different ECMO types.

There was no difference in risk of hemolysis concerning the type of cannulation for either VA- or VV ECMO. On VV ECMO it has already been described that an increase in blood flow may cause mild hemolysis, but even at 3.0–4.5 l/min three-quarters of the fHb values were below 100 mg/l [1,12,16]. A slightly increased hemolysis with higher flow velocity in the cannula of VV ECMO was shown in the literature [12]. Our data confirmed this for VA ECMO when comparing low ($\leq$ 2.5 l/min) vs. high blood flow ($\geq$ 3.0 l/min) in patients with fHb pre $\leq$ 100 mg/l without PHT. Small-sized 15 Fr cannulas and high blood flow ($\geq$ 3.0 l/min) showed higher fHb values compared to 17 Fr cannulas. By matching cannula size and blood flow, the risk of ECMO-induced hemolysis can be controlled [12,30].

Although our ECMO center strives for an optimized ECMO management, significantly less VA ECMO patients survived than on VV ECMO (40% vs. 63%; p$\leq$0.001). Other ECMO centers presented similar survival rates [31,32]. However, the indications for both ECMO types were different.

## Limitations

This study has several limitations. It is a monocentric, retrospective study. Therefore, the reasons for high fHb levels could only be deduced retrospectively from available data. An accurate documentation therefor was decisive. Hemolysis was defined only by determining fHb value; possible sampling errors or transport damage had to be considered (a double determination of the fHb value for conspicuous values served to limit them). Furthermore, it had to be considered that intravascular hemolysis and high fHb values were late markers for blood destruction [12] and can arise for multifactorial reasons.

## Conclusions

VA and VV ECMO patients have many similarities, but more differences. VA ECMO patients presented higher fHb values before ECMO and throughout the ECMO support. VV ECMO, on the other hand, has frequently shown PHTs with increased incidence in Cardiohelp pumps.

The use of different pumps showed no increased risk of hemolysis, independent of ECMO type. Cannulation did not induce hemolysis. The fHb values decreased after ECMO initiation in VA- and VV ECMO.

## Supporting information

**S1 Table. Effect of 15 Fr and 17 Fr inflow cannula during VA ECMO with ECPR on hemolysis.**
(DOCX)

**S2 Table. Effect of the pump type on the frequency of a PHT.**
(DOCX)

**S1 Dataset.**
(XLSX)

## Acknowledgments

We would like to thank all the people who have contributed to the creation and continuous improvement of the database and, of course, the perfusionists and all staff members of the intensive care units of the University Hospital Regensburg, Regensburg, Germany, for their excellent work with these critically ill patients.

## Author Contributions

**Conceptualization:** Alois Philipp, Thomas Mueller, Maik Foltan, Karla Lehle.

**Data curation:** Alois Philipp, Maik Foltan, Matthias Lubnow, Dirk Lunz.

**Formal analysis:** Hannah Appelt, Florian Zeman.

**Investigation:** Thomas Mueller, Matthias Lubnow.

**Methodology:** Hannah Appelt, Alois Philipp, Maik Foltan, Matthias Lubnow, Florian Zeman, Karla Lehle.

**Project administration:** Alois Philipp, Thomas Mueller, Dirk Lunz.

**Resources:** Thomas Mueller.

**Supervision:** Thomas Mueller, Karla Lehle.

**Validation:** Alois Philipp.

**Visualization:** Hannah Appelt, Dirk Lunz.

**Writing – original draft:** Hannah Appelt, Thomas Mueller.

**Writing – review & editing:** Maik Foltan, Matthias Lubnow, Florian Zeman, Karla Lehle.

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
