## [Decision Letter · Decision Letter 0]

8 Oct 2019

PONE-D-19-24231

Factors associated with hemolysis during adult extracorporeal membrane oxygenation (ECMO) – comparison of VA- versus VV ECMO

PLOS ONE

Dear Prof. Lehle,

Thank you for submitting your manuscript to PLOS ONE. After careful consideration, we feel that it has merit but does not fully meet PLOS ONE’s publication criteria as it currently stands. Therefore, we invite you to submit a revised version of the manuscript that addresses the points raised during the review process.

We would appreciate receiving your revised manuscript by Nov 22 2019 11:59PM. To enhance the reproducibility of your results, we recommend that if applicable you deposit your laboratory protocols in protocols.io, where a protocol can be assigned its own identifier (DOI) such that it can be cited independently in the future. For instructions see: http://journals.plos.org/plosone/s/submission-guidelines#loc-laboratory-protocols

We look forward to receiving your revised manuscript.

Kind regards,

Andrea Ballotta

Academic Editor

PLOS ONE

**Journal Requirements:**

2. In the ethics statement in the manuscript and in the online submission form, please provide additional information about the patient records used in your retrospective study, including: a) whether all data were fully anonymized before you accessed them; b) the date range (month and year) during which patients' medical records were accessed; c) the date range (month and year) during which patients whose medical records were selected for this study sought treatment; and d) the source of the medical records analyzed in this work (e.g. hospital, institution or medical center name). If patients provided informed written consent to have data from their medical records used in research, please include this information.

**Additional Editor Comments (if provided):**

First of all thanks to all the authors for this interesting paper dealing with the factors associated with hemolysis during adult V-V and V-A ECMO. On the basis of the reviews i received i deem the manuscript suitable for publication but after major revision only.

**Comments to the Author**

1. Is the manuscript technically sound, and do the data support the conclusions?

Reviewer #1: Partly

Reviewer #2: Yes

2. Has the statistical analysis been performed appropriately and rigorously? 

Reviewer #1: Yes

Reviewer #2: Yes

3. Have the authors made all data underlying the findings in their manuscript fully available?

Reviewer #1: Yes

Reviewer #2: Yes

4. Is the manuscript presented in an intelligible fashion and written in standard English?

Reviewer #1: Yes

Reviewer #2: Yes

5. Review Comments to the Author

Reviewer #1: The article presents the analysis of 1063 patients having undergone ECMO support for different reasons (VA 606 vs VV 457)

The article is interesting and is a hot topic but in my opinion, more details could enrich the article.

- You have to complete the abstract including the years of the study and “consecutive patients” in the Methods section.

- The authors stated the aims of the study was to disclose similarities and differences in VA and VV ECMO associated hemolysis but only in the abstract. In my opinion, it must be clearer also in the main text. Furthermore, the two population are really different in the pre ECMO treatment values in terms of age, BMI, FHb pre, LDH, Bilirubin, CRP pre, and TNF. There was also a difference in age. Even if the authors try to explain these differences probably, a propensity-matched analysis could be useful to better understand the proposed aim and avoid any other bias.

- The references must be order for the number of appearance in a consecutive way.

- Moreover, in my opinion the clinical outcomes could be more important than the simple hemolysis so it could be useful to investigate the adverse clinical events in terms of transfusion and AKI during the ECMO treatment between groups.

- What about the CVVH treatment? Are there any patients that required it during ECMO?

- The differences between VA and VV ECMO in terms of PHT is very interesting but in my opinion a deeper analysis for risk factors could be made even if the duration of ECMO treatment itself maybe is the simplest and easier explanation.

Reviewer #2: Very interesting analysis on hemolysis in ECMO patients, underlying similarities and differences between VV and VA.

A few comments:

- In your analysis ECMO (be it VV or VA) did not seem to promote hemolysis per se, while abnormally high levels of fHb in ECPR and post-cardiotomic VA ECMO seem to be "washed out". Do you believe this is due to a rapid restoration of adequate CO? Possible reasons for high fHb in these subgroups? Liver stasis?

- Was haptoglobin tested?

- How do you define pump head thrombosis? Do you routinely measure fibrinogen and D-dimers? Or is it a clinical/visual diagnosis? Does this prompt a circuit change?

- Smaller cannulae (even if blood flow was not so high) do not seem to increase fHb. Is this because blood flow was reduced (i.e. we generally go up to 2.6 BFI with a 15F)? Please comment.

6. PLOS authors have the option to publish the peer review history of their article (what does this mean?). If published, this will include your full peer review and any attached files.

Reviewer #1: No

Reviewer #2: Yes: Dr. Fabio Sangalli, MD, FASE

---

## [Author Response · Author response to Decision Letter 0]

20 Nov 2019

Reviewer #1: The article presents the analysis of 1063 patients having undergone ECMO support for different reasons (VA 606 vs VV 457)

The article is interesting and is a hot topic but in my opinion, more details could enrich the article.

Comment #1: You have to complete the abstract including the years of the study and “consecutive patients” in the Methods section.

Answer to #1:

Thank you very much for this comment. We included years of the study and the term „consecutive patients“ in the Methods section of the abstract. At the same time, we reformatted the abstract.

Comment #2: The authors stated the aims of the study was to disclose similarities and differences in VA and VV ECMO associated hemolysis but only in the abstract. In my opinion, it must be clearer also in the main text. Furthermore, the two population are really different in the pre ECMO treatment values in terms of age, BMI, FHb pre, LDH, Bilirubin, CRP pre, and TNF. There was also a difference in age. Even if the authors try to explain these differences probably, a propensity-matched analysis could be useful to better understand the proposed aim and avoid any other bias.

Answer to #2:

• In each analysis, we presented the differences / similarities between the two ECMO types (see tables and Figure 2). An exception is the effect of small cannulas on hemolysis. This was published in detail for VV ECMO patients with initial fHb levels ≤ 100 mg/l and 17 Fr cannulas (the most commonly used cannula for VV ECMO). We wanted to avoid a double publication. However, the corresponding reference is already given in the text (Lehle et al. Artificial Organs 2014; 38(5): 391-8).

• The different patient characteristics (Table 2) are one difference and therefore a result. Since these results are important differences that characterize the two study groups, we will not perform a propensity-matched analysis. The decision for a system (VA or VV) does not depend on age, BMI or initial laboratory values - it is a clinical decision. 

• Furthermore, we expanded Table 2 with the proportion of patients with initial acute renal failure and norepinephrine levels (based on your comment below regarding acute kidney injury during ECMO). 

Comment #3: The references must be order for the number of appearance in a consecutive way.

Answer to #3:

Thanks for the note. We have rearranged the order of the references. 

Comment #4: Moreover, in my opinion the clinical outcomes could be more important than the simple hemolysis so it could be useful to investigate the adverse clinical events in terms of transfusion and AKI during the ECMO treatment between groups.

Answer to #4:

According to your comment, we expanded our last result section and entitled this section with “Blood products, ARF and outcome”. This includes a new Table 7. We compared the consumption of blood products per day of ECMO of both groups. There was a significant higher requirement for RBCs, FFPs and PCs during VA ECMO. However, the proportion of patients that required RBCs was comparable. Furthermore, we introduced the events of ARF (acute renal failure) that tended to be higher during VA ECMO. We also discussed these results.

Comment #5: What about the CVVH treatment? Are there any patients that required it during ECMO?

Answer to #5:

As mentioned above (answer to #4), there were patients who needed CVVH treatment during ECMO. However, there was only a trend of more CCVH treatment for VA ECMO compared to VV ECMO (26 % vs. 21 %, p=0.069, respectively). See Table 7.

Comment #6: The differences between VA and VV ECMO in terms of PHT is very interesting but in my opinion a deeper analysis for risk factors could be made even if the duration of ECMO treatment itself maybe is the simplest and easier explanation.

Answer to #6:

We will discuss other risk factors for the occurrence of a PHT:

1. The reason for a PHT could be the formation of a clot on the cannula tip due to long residence time of some cannulas. These clots can detach and aspirate in the pump head. However, there is no evidence for this type of clot accumulation within the pump head. 

2. Anticoagulation regimen was different between VA and VV ECMO. At our center, the target value of aPTT for VV ECMO was 45-50 sec and for VA ECMO 50-60 sec. In particular, patients with worsened ventricular function and intracardiac stasis required higher anticoagulation to prevent a hypercoagulable state and the development of thrombosis. 

3. The consumption of higher RBCs might also be a risk factor for PHT. However, the amount of RBC per ECMO day was significantly higher for VA ECMO compared to VV ECMO.

Reviewer #2: 

Very interesting analysis on hemolysis in ECMO patients, underlying similarities and differences between VV and VA. A few comments:

Comment #1: In your analysis ECMO (be it VV or VA) did not seem to promote hemolysis per se, while abnormally high levels of fHb in ECPR and post-cardiotomic VA ECMO seem to be "washed out". Do you believe this is due to a rapid restoration of adequate CO? Possible reasons for high fHb in these subgroups? Liver stasis?

Answer to #1: 

Excuse our formulation “washed out”. We suspect that the perfusion of the peripheral organs, especially the liver and spleen, is limited in patients with cardiogenic shock or cardiac arrest. Blood stasis within these organs and chest decompression (in about half of the VA ECMO patients – ECPR) may be responsible for destruction of RBCs. A successful and rapid restoration of adequate cardiac output (or organ perfusion) will eliminate the accumulated and toxic fHb in tissue or increase its degradation in spleen and liver.

Reasons for high fHb levels are CPR before ECMO initiation due to RBC damage (Lehle et al. Eur J Heart Failure 2017; 19 (2): 110-6). Furthermore, increased use of RBC transfusions (Vermeulen et al. Critical Care; 2012; 16 (3): R95) which are often needed especially for cardiac surgery, metabolic disorders (Ilani et al. Biochimica et Biophysica Acta; 1990; 1027 (2): 199-204), hypoxia (Foeller et al. IUBMB Life: 2008; 60 (10): 661–668) due to cardiac arrest, various diseases (Rother et al. JAMA; 2005; 293: 1653-1662), bleeding (Vermeulen et al. Critical Care; 2012; 16 (3): R95) and major surgery itself (Vercaemst. J of Extra Corpor Technol 2008; 40 (4): 257-267) may be responsible for high fHb pre levels.

We included these explanations and assumptions in the discussion section.

Comment #2: Was haptoglobin tested?

Answer to #2: 

No. Haptoglobin testing was not performed at our ECMO center. Haptoglobin is a Hb scavenger and normally cleared high levels of Hb. However, this clearing mechanism becomes saturated and exhasted in case of excessive hemolysis – in particular for patients with initial levels of fHb.

(Vercaemst. J of Extra Corpor Technol 2008; 40 (4): 257-267). We did not include this comment in our manuscript. 

Comment #3: How do you define pump head thrombosis? Do you routinely measure fibrinogen and D-dimers? Or is it a clinical/visual diagnosis? Does this prompt a circuit change?

Answer to #3: 

• Pump head thrombosis is defined as a rapid and substantial increase in fHb (>300 mg/l) accompanied mostly by a rapid decrease in platelets and an abnormal noise/vibration of pump head. After circuit change the levels of fHb decreased and platelets normalized (Lubnow et al. 2014; PloS one 9 (12): 4112316). In these cases there was always a clot within the removed pump head. 

• We routinely determine some coagulation factors such as fibrinogen, D-Dimers, platelets once a day. However, neither D-Dimers nor fibrinogen has any significance for the detection of a pump head thrombosis. This was described in detail in the study Lubnow et al. 2014; PloS one 9 (12): 4112316

• PHT as an acute event is always associated with an immediate change of system.

• We included the definition of a pump head thrombosis within the text including the reference. 

Comment #4: Smaller cannulae (even if blood flow was not so high) do not seem to increase fHb. Is this because blood flow was reduced (i.e. we generally go up to 2.6 BFI with a 15F)? Please comment.

Answer to #4:

Yes, of course. A higher blood flow (BF) through a small-sized cannula (with a small diameter/radius) will increase flow velocity (FV). 

BF = FV * Area of the cannula = FV * r2 *π

An increase in FV also increased shear rates and may increase the risk of RBC damage.

In the special case of VA ECMO, the usage of a 15 Fr cannula – the most commonly used cannula for VA ECMO - showed increased hemolysis at blood flow rates ≥3 l/min. The decision for a cannula depends on the clinical need for respiratory support of the patient. Sometimes the patient needs more at ECMO than initially thought. An exchange of the cannula (with a larger diameter) is avoided. Instead, the blood flow will be increased, which is associated with an increased risk of hemolysis as shown in the present study.

We included this comment in the discussion section.

---

## [Decision Letter · Decision Letter 1]

31 Dec 2019

Factors associated with hemolysis during extracorporeal membrane oxygenation (ECMO) – Comparison of VA- versus VV ECMO

PONE-D-19-24231R1

Dear Dr. Lehle,

We are pleased to inform you that your manuscript has been judged scientifically suitable for publication and will be formally accepted for publication once it complies with all outstanding technical requirements.

With kind regards,

Andrea Ballotta

Academic Editor

PLOS ONE

Additional Editor Comments (optional):

Thank you for all your comments. in the basis of the reviewers' evaluation i deemed the paper suitable for publication.

Reviewers' comments:

Reviewer's Responses to Questions

**Comments to the Author**

1. If the authors have adequately addressed your comments raised in a previous round of review and you feel that this manuscript is now acceptable for publication, you may indicate that here to bypass the “Comments to the Author” section, enter your conflict of interest statement in the “Confidential to Editor” section, and submit your "Accept" recommendation.

Reviewer #1: All comments have been addressed

Reviewer #2: All comments have been addressed

2. Is the manuscript technically sound, and do the data support the conclusions?

Reviewer #1: Yes

Reviewer #2: Yes

3. Has the statistical analysis been performed appropriately and rigorously? 

Reviewer #1: Yes

Reviewer #2: Yes

4. Have the authors made all data underlying the findings in their manuscript fully available?

Reviewer #1: Yes

Reviewer #2: Yes

5. Is the manuscript presented in an intelligible fashion and written in standard English?

Reviewer #1: Yes

Reviewer #2: Yes

6. Review Comments to the Author

Reviewer #1: (No Response)

Reviewer #2: All comments were addressed by the Authors and the paper was edited accordingly. This reviewer has no further comments.

7. PLOS authors have the option to publish the peer review history of their article (what does this mean?). If published, this will include your full peer review and any attached files.

Reviewer #1: No

Reviewer #2: Yes: Dr. Fabio Sangalli, MD, FASE

---

## [Editor Report · Acceptance letter]

10 Jan 2020

PONE-D-19-24231R1 

Factors associated with hemolysis during extracorporeal membrane oxygenation (ECMO) – Comparison of VA- versus VV ECMO 

Dear Dr. Lehle:

I am pleased to inform you that your manuscript has been deemed suitable for publication in PLOS ONE. Congratulations! Your manuscript is now with our production department. 

With kind regards,

on behalf of

Dr. Andrea Ballotta 

Academic Editor

PLOS ONE